# Leptin-Responsive MiR-4443 Is a Small Regulatory RNA Independent of the Canonic MicroRNA Biogenesis Pathway

**DOI:** 10.3390/biom10020293

**Published:** 2020-02-13

**Authors:** Ari Meerson

**Affiliations:** 1MIGAL—Galilee Research Institute, POB 831, Kiryat Shmona 1101602, Israel; arim@migal.org.il; Tel.: +972-4-695-5022; 2Faculty of Sciences, Tel-Hai Academic College, Upper Galilee 1220800, Israel

**Keywords:** noncoding RNA, microRNA, biogenesis, Drosha, Dicer, Exportin, microprocessor, non-canonical, leptin, insulin

## Abstract

The human small RNA miR-4443 is functionally involved in several types of cancer and in the biology of the immune system, downstream of insulin and leptin signaling. Next generation sequencing evidence and structural prediction suggest that miR-4443 is not produced via the canonical Drosha–Exportin 5–Dicer pathway of microRNA biogenesis. We tested this hypothesis by using qRT-PCR to measure miR-4443 and other microRNA levels in HCT-116 cells with Drosha, Exportin 5, and Dicer knockouts, as well as in the parental cell line. Neither of the knockouts decreased miR-4443 levels, while the levels of canonical microRNAs (miR-21 and let-7f-5p) were dramatically reduced. Previously published Ago2-RIP-Seq data suggest a limited incorporation of miR-4443 into RISC, in agreement with the functional studies. The miR-4443 locus shows conservation in primates but not in other mammals, while its seed region appears in additional microRNAs. Our results suggest that miR-4443 is a Drosha, Exportin 5, and Dicer-independent, non-canonical small RNA produced by a yet unknown biogenesis pathway.

## 1. Introduction

The human small RNA miR-4443 (miRbase accession MI0016786, http://www.mirbase.org/cgi-bin/mirna_entry.pl?acc=MI0016786) is functionally involved in several types of cancer and in the biology of the immune system. We previously reported that in cultured colon cancer cells, miR-4443 was upregulated by leptin and insulin in a MEK1/2-dependent manner, decreased invasion and proliferation, and directly downregulated pro-metastatic NCOA1 and TRAF4. Insulin and/or leptin resistance (e.g., in obesity) may neutralize this tumor-suppressive pathway, increasing the risk of developing cancer [1]. Supporting this notion, the miR-4443 locus is frequently deleted in cancers based on TCGA data; and a tumor suppressor role for miR-4443 was also reported in other cancer types, i.e., osteosarcoma [2], hepatocellular carcinoma [3], ovarian cancer [4], and glioblastoma [5]. On the other hand, miR-4443 was shown to promote the resistance of non-small cell lung cancer cells, as well as metastatic breast cancer cells, to the chemotherapeutic agent epirubicin [6,7]; and its elevated levels in plasma were suggested as a biomarker of glial tumors [8]. In addition to its function in cancer cells, miR-4443 was reported to regulate the behavior of immune cells, i.e., CD4+ T cells in the context of Graves’ Disease [9] as well as mast cells, notably via microvesicles secreted from T cells [10]. Finally, a recent study reported that transfection with anti-miR-4443 oligos modified the response of differentiated neuronal cells (SH-SY5Y) to oxidative stress [11]. miR-4443 differs from most human microRNAs in the following aspects (summarized in Figure 1):

(a) The most common isoform of the mature sequence is only 17 nucleotides long. In comparison, the median length of mature human microRNAs is 22 nucleotides. A prior study of microRNA length distribution found that 60.7% of the cancer-associated human microRNAs analyzed were 22 nucleotides long, while 99.4% fell within the 22 ± 2 nucleotide bracket [12];

(b) The predicted terminal loop of pre-miR-4443 is only four nucleotides long. It is well known that the size and structural flexibility of the microRNA precursor terminal loop are important for both Drosha and Dicer-mediated processing [13,14], and to a lesser degree for Exportin 5 binding [15];

(c) Deep sequencing evidence from ~23 K reads shows only the 5′ mature miR-4443 product and no passenger strand product from pre-miR-4443.

Combined, these properties exclude miR-4443 from stringent microRNA annotation criteria, e.g., the MirGeneDB database [16] (http://mirgenedb.org/) and suggest that miR-4443 is not a product of the canonical Drosha–Exportin 5–Dicer microRNA biogenesis pathway.

## 2. Materials and Methods

### 2.1. Cells and Cell Culture

The following HCT-116 parental strain and knockout strains, originally developed and described by the lab of Narry Kim [17], were obtained from the Korean Collection for Type Cultures (KCTC, Jeollabuk-do, Korea) in January 2019, and kept frozen in liquid nitrogen until use. These included Drosha knockout (KO) clone #40, Exportin 5 KO clones #19 and #19-1, and Dicer KO clones #43 and #45. Cells were cultured based on ATCC recommendations. Culture media and fetal bovine serum were obtained from Biological Industries (Israel). All strains were grown in triplicates in Greiner Bio-One (Kremsmünster, Austria) Cellstar 6-well tissue culture plates until reaching ~80% confluence.

### 2.2. RNA Isolation

Isolation of total RNA (including miRNAs) was carried out using the Qiagen (Venlo, Netherlands) miRNeasy Kit according to the manufacturer’s instructions. RNA concentration was assessed using a Thermo-Fisher Scientific (Waltham, MA, USA) NanoDrop 8000 Spectrophotometer. All RNA samples had an OD260/280 of ≥1.8. A total of 1 μg of RNA from each sample was used for qRT-PCR.

### 2.3. qRT-PCR

Reverse transcription, primer design, and quantitative PCR were performed using SYBR Green chemistry (iTaq SYBR Master Mix, BioRad, Hercules, CA, USA) and DNA primers (Sigma-Aldrich, St. Louis, MO, USA). Primer extension was used for miRNA quantification as previously described [1,18,19]. This qRT-PCR method is highly accurate and reproducible [18,19], and was chosen over hydrolysis probe chemistry due to its lower cost. The sequence of the RT-primer was 5′-CAGGTCCAGTTTTTTTTTTTTTTTVN-3′, where V is A, C and G and N is A, C, G, and T. qPCR primer sequences are provided in Table 1. All primers were tested for efficiency (by serial dilutions) and specificity (by melting peak analysis). RT was performed on an Applied Biosystems ABI-9600 (Thermo-Fisher Scientific) with reagents from New England Biolabs (Ipswich, MA, USA) as previously described [1]. qPCR was performed in technical quadruplicates on an Applied Biosystems ABI-7900HT Sequence Detection System (Thermo-Fisher Scientific) equipped with a 384-well block. Data were analyzed using the SDS 2.3 software (Applied Biosystems, Thermo-Fisher Scientific) and Microsoft Excel. Relative quantification and the ΔCq method were used.

### 2.4. Bioinformatics and Online Datasets

Raw read files from Krell et al. [20] were obtained from the European Nucleotide Archive, projects PRJEB3396 and PRJEB3157 for PAR-CLIP-seq of AGO2 binding sites and AGO-RIP-seq data, respectively. MiR-4443 locus conservation was assessed using NCBI BLAST [21], human reference genome release GRCh38, and the NCBI genome browser. For seed region comparison, microRNA sequences were retrieved from miRBase, release 22.1 [22], and scanned for identification of all miRNAs with the same seed sequence (UGGAGG) at positions 2–7.

### 2.5. Statistics

For statistical tests, Student’s *t*-test was used.

## 3. Results and Discussion

To assess whether miR-4443 biogenesis is canonical, we used qRT-PCR to measure miR-4443 levels in cultured HCT-116 colon cancer cell lines with Drosha, Exportin 5, and Dicer knockouts [17], as well as in the parental cell line. We also measured the levels of two canonical microRNAs, miR-21 and let-7f-5p, in the same cells. As expected, the levels of both canonical microRNAs (miR-21 and let-7f-5p) were dramatically reduced in the knockout lines (Figure 2). Thus, Drosha, Exportin 5, and Dicer knockouts caused ~100×, ~3× and ~10× decreases, respectively, in the levels of both miR-21 and let-7f-5p. All these decreases (except for the decrease in let-7f-5p in one of the Exportin 5 KO lines) were statistically significant (*p* < 0.05). In contrast, neither of the knockouts decreased miR-4443 levels (Figure 2). Indeed, miR-4443 levels were between 2× and 7× higher in the knockout lines than in the parental line, although this difference was not statistically significant and may have resulted from differences in total small RNA quantity between the samples.

As these results raised the possibility that miR-4443 is not an actual microRNA (functionally defined by its association with RISC proteins), we examined the evidence of its association with Argonaute 2 (Ago2) in published Ago2-RIP-Seq datasets. Utilizing the raw data from a study by Krell et al. [20], notably also performed in HCT-116 cells, we identified the core sequence of miR-4443 (TGGAGGCGTGGGTT) in both input and Ago2-RIP-Seq reads, as well as in the PAR-CLIP reads. The frequency of this sequence was low (~1/10^5^ of total reads, and ~1/10^3^ compared to common microRNAs like miR-21) in both the input and the Ago2-RIP-Seq data, which does not necessarily indicate weak binding to Ago2 due to the low overall cytoplasmic levels of this RNA (e.g., Figure 1B).

To assess the cross-species conservation of miR-4443, we examined the conservation of the human *miR-4443* genome locus. Additionally, we searched for other annotated microRNAs with the same seed region as miR-4443, defined as a contiguous string of at least six nucleotides beginning at position two [23]. While no significant cross-vertebrate or cross-mammal conservation was observed for the *miR-4443* locus, it was conserved in primates (Figure 3), albeit the similar sequences in other primates were not annotated as microRNAs. An identical seed sequence was identified in additional microRNAs of human, mouse, rat, and bovine origin, as well as in non-mammalian sequences (Table 2). While seed homology is not an indicator of miR-4443′s relatedness to other microRNAs (due to the high probability of a chance match), it does suggest that this RNA is likely to share a set of targets with other microRNAs.

Although some researchers have suggested excluding non-canonical microRNAs from being considered microRNAs at all, many examples of these molecules are known, and some clearly have important biological functions. Most non-canonical microRNAs require Dicer for their final processing step [24]; however, there are notable exceptions. One well-known example is miR-451a, which is abundantly expressed in erythrocytes, plays an important role in erythropoiesis [25,26], and shows evolutionary conservation in vertebrates [27]. miR-451a biogenesis has been shown to be mediated by Argonaute 2, independent of Dicer but not of Drosha [25,27,28]. Another class of non-canonical microRNAs, named “simtrons”, which includes miR-1225 and miR-1228, also require Drosha but not Dicer or Exportin 5 for their processing [29]. Our results suggest that miR-4443 represents yet another type of small regulatory RNAs, independent of Drosha, Exportin 5, and Dicer, and produced by a hitherto unknown biogenesis pathway.

Despite this divergent biogenesis of miR-4443, we have found evidence of its binding to Argonaute 2. Additionally, a growing number of functional studies suggest that miR-4443 regulates the expression of target genes by binding the 3′ UTRs of their mRNA. Although more conclusive evidence of RISC incorporation of miR-4443 is necessary, existing evidence (in this and other studies) suggests that miR-4443 functions in RISC-mediated gene regulation. Its inclusion or exclusion from microRNA nomenclature is therefore subject to opinion of what constitutes a microRNA. Furthermore, it seems that, due to the several known non-canonically produced microRNAs, it is indeed more acceptable to draw the line based on function rather than on biogenesis.

## 4. Conclusions

Our results suggest that miR-4443 is a small regulatory RNA produced by a hitherto unknown biogenesis pathway. The physiologically and pathologically relevant functions of miR-4443 warrant a better understanding of this molecule, its upstream regulation, and downstream effectors. Further experiments are required to elucidate the biogenesis of miR-4443, and to address the question whether it is unique or indeed representative of a novel class of small regulatory RNAs.

## Figures and Tables

**Figure 1 biomolecules-10-00293-f001:**
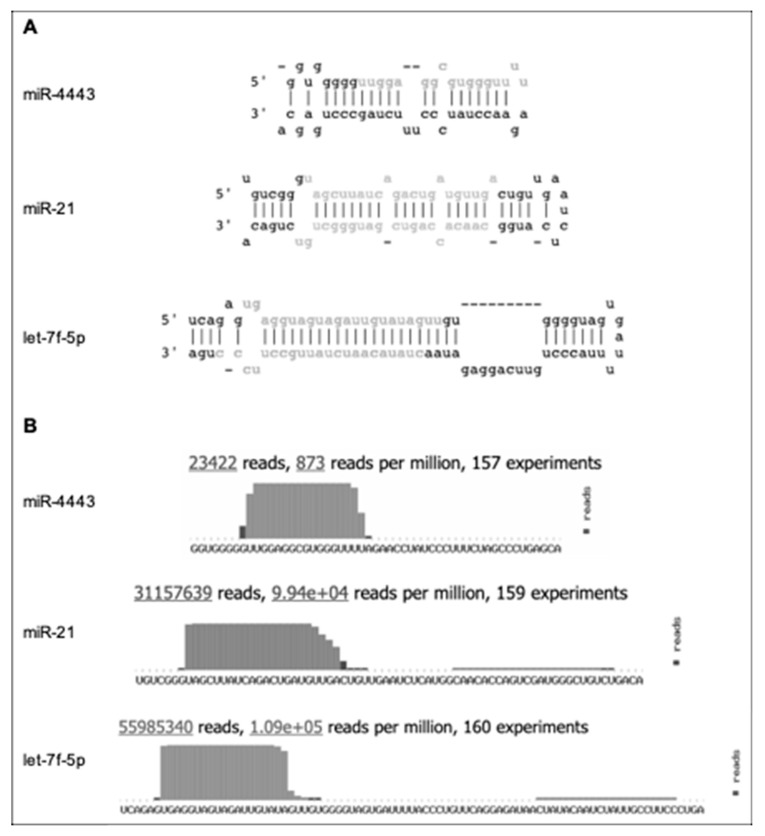
Indirect evidence for non-canonical biogenesis of miR-4443. Adapted from miRbase (http://www.mirbase.org/cgi-bin/mirna_entry.pl?acc=MI0016786), with let-7f-5p and miR-21 included for comparison. (**A**) The predicted stem-loop structure of pre-miR-4443 compared with the structures of pre-let-7f-5p and pre-miR-21. Mature microRNA sequences are shown in gray. (**B**) The deep sequencing evidence for mature miR-4443, compared with let-7f-5p and miR-21. Note the short (17 nt) mature sequence, short terminal loop, and lack of evidence for passenger strand for miR-4443.

**Figure 2 biomolecules-10-00293-f002:**
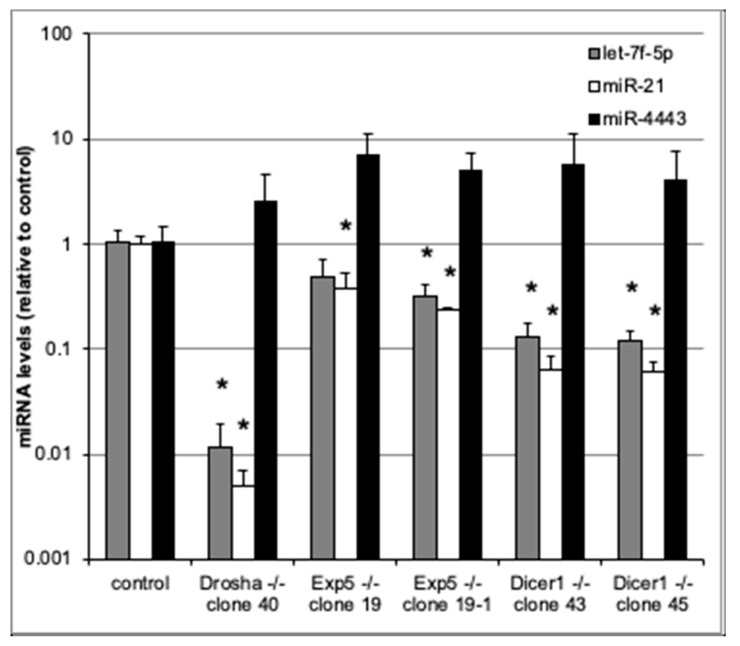
qRT-PCR for miR-4443, let7f-5p, and miR-21 in cultured HCT-116 cells with knockouts of Drosha, Exportin 5, or Dicer genes and controls. The levels of each microRNA relative to control are shown. Note logarithmic scale of vertical axis. Bars: st. dev. from biological triplicates. * *p* < 0.05 (*t* test).

**Figure 3 biomolecules-10-00293-f003:**
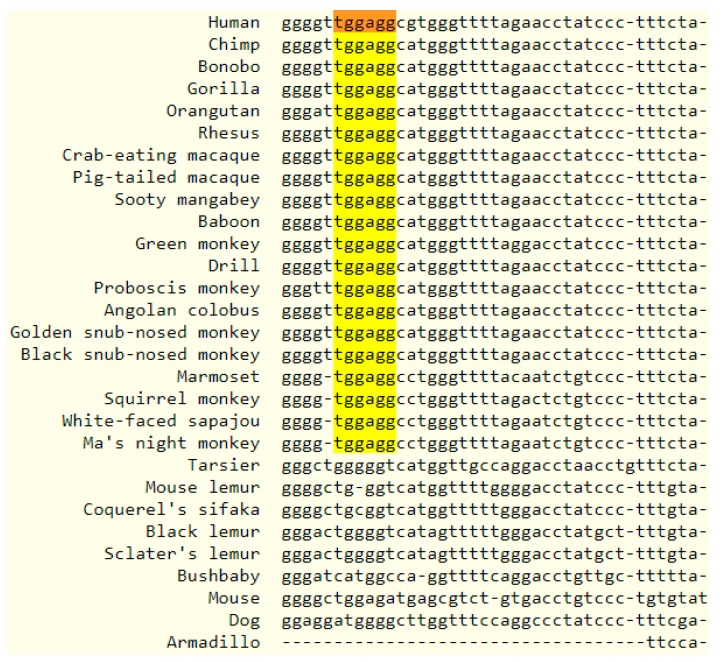
Multiple alignment output showing the human *mir-4443* locus and similar loci in 27 primates and 3 other mammals. The identical 6-nucleotide seed region is highlighted.

**Table 1 biomolecules-10-00293-t001:** qPCR primers used (see [1,18,19]).

	Forward Primer	Reverse Primer
**miR-4443**	GTTGGAGGCGTGGGT	GGTCCAGTTTTTTTTTTTTTTTAAAACC
**let-7f-5p**	CGCAGTGAGGTAGTAGATTG	GGTCCAGTTTTTTTTTTTTTTTAACTATAC
**miR-21-1**	GCAGTAGCTTATCAGACTGATG	GGTCCAGTTTTTTTTTTTTTTTCAAC

**Table 2 biomolecules-10-00293-t002:** Animal miRbase-annotated microRNAs sharing a 6-nucleotide seed (in bold) with miR-4443.

	Organism	miRNA Name	Sequence	5p/3p
1	Homo sapiens	hsa-miR-4443	U**UGGAGG**CGUGGGUUUU	5p
2	Homo sapiens	hsa-miR-6515-5p	U**UGGAGG**GUGUGGAAGACAUC	5p
3	Mus musculus	mmu-miR-1843b-5p	A**UGGAGG**UCUCUGUCUGACUU	5p
4	Mus musculus	mmu-miR-6982-5p	C**UGGAGG**AUCGCAGGGGUGGCCUGG	5p
5	Rattus norvegicus	rno-miR-1843b-5p	A**UGGAGG**UCUCUGUCUGACUUAG	5p
6	Bos taurus	bta-miR-2893	G**UGGAGG**AGAAUGCCCGGGG	5p
7	Bos taurus	bta-miR-12054	C**UGGAGG**UGGGGAUGCAC	3p
8	Gallus gallus	gga-miR-3532-3p	U**UGGAGG**CUGCAGUGUCAUGGU	3p
9	Echinococcus granulosus	egr-miR-10242-3p	G**UGGAGG**CCAUCCAAGUAGC	3p

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
