# Peer review of "Leptin-Responsive MiR-4443 Is a Small Regulatory RNA Independent of the Canonic MicroRNA Biogenesis Pathway"

_biomolecules, 2020, doi:10.3390/biom10020293_

Round 1
Reviewer 1 Report
The manuscript entitled "Leptin-responsive miR-4443 is a small regulatory 2 RNA independent of the canonic microRNA 3 biogenesis pathway" is very interesting and good written.
Comment:
The discussion is short and should be extendedAuthor Response
I’d like to thank all the reviewers for their helpful comments. The manuscript has now been revised to address their concerns. A point-by-point response follows (the reviewers’ comments are in italics).
Reviewer 1:
The manuscript entitled "Leptin-responsive miR-4443 is a small regulatory RNA independent of the canonic microRNA biogenesis pathway" is very interesting and good written.
I thank the reviewer for the positive assessment.
Comment: The discussion is short and should be extended
While it wasn’t my intention to deviate from the Short Communication format, the discussion section was expanded to address this comment
Reviewer 2 Report
The author want to provide that miR-4443 is not synthesized by canonic microRNA biogenesis pathway. But the only supportive evidence is figure 2 by knockout of Drosha, Exportin 5 or Dicer. The repression of the controls Let-7f and miR-21 in most cases are not very significant. Worst still, there is no date (such as Western blotting) to support that the cells did not express any Drosha, Exportin 5 or Dicer.
Author Response
I’d like to thank all the reviewers for their helpful comments. The manuscript has now been revised to address their concerns. A point-by-point response follows (the reviewers’ comments are in italics).
Reviewer 2:
The author want to provide that miR-4443 is not synthesized by canonic microRNA biogenesis pathway. But the only supportive evidence is figure 2 by knockout of Drosha, Exportin 5 or Dicer.
Drosha, Exportin 5 and Dicer are all critical components in the canonical processing of microRNAs. The fact that miR-4443 levels were unaffected by the knockout of any of these genes, is clear cut evidence that miR-4443 does not go through canonical microRNA biogenesis.
The repression of the controls Let-7f and miR-21 in most cases are not very significant.
As noted in the manuscript, Drosha, Exportin 5 and Dicer knockouts caused ~100X, ~3X and ~10X decreases respectively, in the levels of both miR-21 and let-7f-5p. All these decreases (except for the decrease in let-7f-5p in one of the Exportin 5 KO lines) were statistically significant (p<0.05). These results are in agreement with previously published findings, for example Kim et al., PNAS 2016.
Worst still, there is no date (such as Western blotting) to support that the cells did not express any Drosha, Exportin 5 or Dicer.
The knockout cell lines used in the experiment were previously described in Kim et al., PNAS 2016 as referenced in the manuscript. In that paper, Western blot results that confirm the knockout of Drosha, Exportin 5 and Dicer in these cell lines are presented in Fig. 1C. To address the reviewer’s comment, the relevant figure panel from Kim et al. now appears in the revised manuscript as Fig. 2A with the appropriate credit given.
Reviewer 3 Report
In this Manuscript, Ari Meerson sets out to elucidate the biogenesis mechanism of miR-4443, a leptin-responsive small RNA with a tumor suppressor role.
Using an elegant and well controlled approach, the author determines that miR-4443 1) is not processed by Drosha, 2) it is not processed by Dicer, 3) it is not shuttled to the cytosol by Exportin5, 4) the mature product is 17 nt rather than the regular 22 nt, and 5) it is not likely loaded into Ago. The authors use the bonafide miRNAs let-7 and miR-21 as processing controls.
Cross-species comparison rendered that human miR-4443 is mainly conserved in primates, albeit not annotated as miRNA. The specific search for other miRNAs sharing the same seed as miR-4443 uncovered 8 miRNAs across vertebrates with the same seed as miR-4443.
The author concludes that miR-4443 is a small regulatory RNA process by an unknown mechanism independent of all the miRNA machinery.
While I commend the author for the experimental approach and agree with the conclusion that miR-4443 may represent a novel class of small regulatory RNA, I consider that this manuscript can cause confusion to the non-miRNA expert readers as they can still consider that miR-4443 is a miRNA processed by a non-canonical pathway. In my opinion, miR-4443 is not at all a miRNA and should be removed from the miRNA database. The author does not claim that it is a miRNA, but by merely using miR-4443 across the manuscript and looking for seed homology causes confusion. Therefore, I consider that the manuscript is not suitable for publication in Biomolecules in the current form.
Major concern:
1) The title and the body of the manuscript can cause confusion ad lead to interpret that miR-4443 is a non-canonical miRNA. miR-4443 it is not a miRNA in any shape or form, as the author well controlled processing experiment demonstrates. The author should rewrite the manuscript to avoid any confusion and explicitly clarify that this work is an experimental validation that miR-4443 is not a miRNA, should have a different name, and should be removed from miRbase.
2) Since miR-4443 is not a miRNA, the seed homology comparison is not appropriate and should be removed. The fact that there are other sequences in miRbase with the same seed is little informative, as one every 4096 molecules will have the same seed just by chance. Currently there are 38589 entries in miRbase, which by chance should produce 9 sequences with the miR-4443 seed.
Author Response
I’d like to thank all the reviewers for their helpful comments. The manuscript has now been revised to address their concerns. A point-by-point response follows (the reviewers’ comments are in italics).
Reviewer 3:
…While I commend the author for the experimental approach and agree with the conclusion that miR-4443 may represent a novel class of small regulatory RNA, I consider that this manuscript can cause confusion to the non-miRNA expert readers as they can still consider that miR-4443 is a miRNA processed by a non-canonical pathway. In my opinion, miR-4443 is not at all a miRNA and should be removed from the miRNA database. The author does not claim that it is a miRNA, but by merely using miR-4443 across the manuscript and looking for seed homology causes confusion…
…1) The title and the body of the manuscript can cause confusion ad lead to interpret that miR-4443 is a non-canonical miRNA. miR-4443 it is not a miRNA in any shape or form, as the author well controlled processing experiment demonstrates. The author should rewrite the manuscript to avoid any confusion and explicitly clarify that this work is an experimental validation that miR-4443 is not a miRNA, should have a different name, and should be removed from miRbase.
First of all, I would like to thank the reviewer for the in-depth assessment and the favorable comments. I also agree that nomenclature issues may be confusing, and am committed to address this confusion in the revised version of the manuscript. I believe that the following points have to be considered:
1) Although the processing of miR-4443 is indeed independent of Drosha, Exportin 5 and Dicer, we have found evidence of its binding to Argonaute 2, as indicated in the manuscript.
2) Additionally, a growing number of functional studies suggest that miR-4443 regulates the expression of target genes by binding the 3’ UTRs of their mRNA.
Given these two points, miR-4443 likely functions as a microRNA in RISC-mediated gene regulation, despite its divergent biogenesis. Its inclusion or exclusion from microRNA nomenclature is therefore subject to opinion of what constitutes a microRNA. Furthermore, it seems that, given the several known non-canonically produced microRNAs, it is indeed more acceptable to draw the line based on function and not on biogenesis. As more conclusive evidence of RISC incorporation of miR-4443 is necessary, I cannot insist on miR-4443 being a true microRNA; but I believe enough evidence exists (in this and other studies) to at least retain that possibility.
2) Since miR-4443 is not a miRNA, the seed homology comparison is not appropriate and should be removed. The fact that there are other sequences in miRbase with the same seed is little informative, as one every 4096 molecules will have the same seed just by chance. Currently there are 38589 entries in miRbase, which by chance should produce 9 sequences with the miR-4443 seed.
I agree that seed homology is not an indicator of miR-4443’s relatedness to other microRNAs in miRbase. The point of this analysis, however, is to show, based on its seed region which is essential for target binding, that this RNA is likely to share a set of targets with other microRNAs.
In the revised manuscript, I have expanded the Discussion part to address the main points raised by Reviewer 3. My hope is that the revised manuscript addresses the reviewers’ comments, and feel that the manuscript has been improved by their critique.
Round 2
Reviewer 2 Report
Did not find point to point responses.
Author Response
I’d like to thank all the reviewers for their helpful comments. The manuscript has now been revised to address their concerns. A point-by-point response follows (the reviewers’ comments are in italics). (As indicated in my e-mail from 08/02/02 to Marija Dragojevic, this response was already given after Round 1 of review, and can be seen there).
Reviewer 2:
The author want to provide that miR-4443 is not synthesized by canonic microRNA biogenesis pathway. But the only supportive evidence is figure 2 by knockout of Drosha, Exportin 5 or Dicer.
Drosha, Exportin 5 and Dicer are all critical components in the canonical processing of microRNAs. The fact that miR-4443 levels were unaffected by the knockout of any of these genes, is clear cut evidence that miR-4443 does not go through canonical microRNA biogenesis.
The repression of the controls Let-7f and miR-21 in most cases are not very significant.
As noted in the manuscript, Drosha, Exportin 5 and Dicer knockouts caused ~100X, ~3X and ~10X decreases respectively, in the levels of both miR-21 and let-7f-5p. All these decreases (except for the decrease in let-7f-5p in one of the Exportin 5 KO lines) were statistically significant (p<0.05). These results are in agreement with previously published findings, for example Kim et al., PNAS 2016.
Worst still, there is no date (such as Western blotting) to support that the cells did not express any Drosha, Exportin 5 or Dicer.
The knockout cell lines used in the experiment were previously described in Kim et al., PNAS 2016 as referenced in the manuscript. In that paper, Western blot results that confirm the knockout of Drosha, Exportin 5 and Dicer in these cell lines are presented in Fig. 1C.